# "If it is not made easy for me, I will just not bother". A qualitative exploration of the barriers and facilitators to recycling plastics

**Deborah Roy**[1][☯], **Emma Berry**[2][☯]*, **Martin Dempster**[2][☯]

**1** Department of Psychology, University of Bath, Bath, England, **2** School of Psychology, Queens University Belfast, Belfast, Northern Ireland

☯ These authors contributed equally to this work.
* e.berry@qub.ac.uk

**Data Availability Statement:** The dataset is held in a secure repository at Queens University Belfast Psychology Department. A Supplementary Data File containing a compilation of further supporting

## Abstract

Despite significant investment to increase recycling facilities and kerbside collection of waste materials, plastic packaging is frequently discarded as litter, resulting in significant environmental harm. This research uses qualitative methods to explore the contextual and psychological factors that influence plastic waste disposal behaviour from the perspectives of consumers. This research also reports key results from a brief online survey exploring consumer perspectives toward plastics and plastic recycling. A total of N = 18 adults living in Northern Ireland (NI) participated in a semi-structured interview and N = 756 adults living in NI took part in an online survey. Interview data was analysed via a semi-directed content analysis approach, using the COM-B behaviour change model as a guiding framework. Survey data underwent descriptive and frequency analysis. Collectively, the findings suggest that environmental concern exists among consumers generally, but there is a degree of ambivalence toward recycling that reflects a gap between *intentions* to recycle and actual recycling behaviour. Plastic recycling behaviour is hindered by three common barriers: 1. confusion and uncertainty about which plastic materials can be recycled (exacerbated by the abundance of plastic products available) 2. perceiving plastic recycling to be less of a personal priority in daily life 3. perceiving that local government and manufacturers have a responsibility to make plastic recycling easier. As recycling is simply not a priority for many individuals, efforts should instead be placed on providing greater scaffolding to make the process of recycling less tedious, confusing, and more habitual. Visual cues on product packing and recycling resources can address ambiguity about which plastic materials can/cannot be recycled and increasing opportunities to recycle (via consistent availability of recycling bins) can reduce the physical burden of accessing recycling resources. Such interventions, based on environmental restructuring and enablement, may increase motivations to recycle by reducing the cognitive and physical burden of recycling, supporting healthier recycling habits.

quotes are published with the manuscript. Interested researchers wishing further data should initially contact Professor Martin Dempster, at Queen's University, Belfast. Minimal datasets to support study replication and preliminary analysis are available in the supplementary documents uploaded.

**Funding:** This work was supported by the Engineering and Physical Sciences Research Council (ESPRC) Grant number EP/S025545/1. https://epsrc.ukri.org/ The funders had no role in study design, data collection and analysis, decision to publish, or preparation of the manuscript.

**Competing interests:** The authors have declared that no competing interests exist.

# Introduction

Plastic materials have become a valuable resource across the world and used by many industries [1]. Materials such as Polyvinyl chloride (PVS) are used to insulate homes, Polyethylene terephthalate (PET) is used to package food to keep it fresh, and Polypropylene (PE) is used to bottle water and drinks [2]. Never has the benefits of plastic been more visible than in the production of Protective Equipment (PPE) to prevent the spread of Covid-19. Plastic packaging was developed to address a sustainability issue, to support the longevity of products. But today the overwhelming amount of plastic products surpasses our ability to manage the environmental harm of discarded plastic. Visible in our rivers and beaches, plastic items have now become a symbol of pollution and environmental harm caused by humans [3].

One important action, supported globally, but more recently by America's leading plastic manufacturers [1] is to "*engage the entire plastics value chain, from plastic makers to brand companies, to all Americans*". The vision being, that plastics are routinely reused, and plastic waste is diverted from landfills and our oceans. In Europe, laudable goals have been set to increase the reuse of plastic waste, and yet the production of new or virgin plastics, far exceeds recycling rates [4]. While the reduction and reuse of plastics is a higher level preventative measure to plastic waste, it is recognised that this is not feasible for all plastic materials and many plastic packaging, if appropriately recycled, can be reprocessed into other economically valuable materials. However at present, approximately half of the annual global production of solid plastics, or 150 million tonnes, is thrown away worldwide each year [5]. Inappropriate disposal of potentially recyclable material means that plastics often reside in landfill [6]. This is a missed economic, and potentially energy-saving, opportunity [5]. Increasing the reuse of household plastic waste is a prime example of how we can mitigate the harm of plastic waste and generate profit through used packaging. But this is largely determined by consumer behaviour. For clarity, we define (plastic) recycling behaviour as the act of disposing of used/unwanted plastic packaging in a bin or other resource which is intended to recover that packaging for reprocessing.

## The role of behaviour change theory

To date, most research exploring consumer recycling behaviour has focused on understanding influences on human motivation and a dominant theory has been the Theory of Planned Behaviour (TPB) [7, 8]. Despite its widespread application, a major review of the use of TPB by Yurieva and colleagues (2020) revealed that 86% of the research studies in their review, focus on factors which influence attitudes (indirect) rather than factors which influence recycling behaviours directly [9]. These indirect factors include demographics [10–13], rewards [12], feedback [14], recycling scheme design [15], scheme knowledge [16], environmental concern [17], antecedent behaviour [18], personal norms [19], and emotions [20, 21]. Deeply entrenched habits (of not recycling) also sustain low rates of recycling across consumer households [22]. The reality is that there are multiple factors that influence our attitudes towards recycling plastics and successful behaviour change also invariably relies upon context. A theoretical model is therefore needed which can capture the multi-faceted predictors of recycling behaviour is needed. The Capability, Opportunity and Motivation (COM-B) model of behaviour [23, 24] is a more pluralistic framework that considers situational factors such as resource and context, as well as psychological factors such as knowledge, skills, and benefit vs. consequence beliefs. The COM-B hypothesises that the interaction occurs between three components that influence and therefore have a role to play in behaviour change. These are; Capability, Opportunity and Motivation (COM), causes the performance (or not) of Behaviour (B) [23, 24]. Capability refers to psychological and physical capability, where psychological

capability is the capacity to engage in the thought processes necessary to create the intent toperformthe behaviour, and physical capability is the capacity to engage in the physical activity necessary for performance of the behaviour. Opportunity covers all those factors that lie outside the individual that make the behaviour possible and can prompt it. It is divided into physical opportunity (the opportunities provided by the environment such as kerbside collection services and recycling bins) and social opportunity (the impact of the cultural context on our thoughts/beliefs) [23, 24]. Motivation is divided into reflective and automatic motivation. Reflective motivation concerns our evaluations/beliefs that are relevant to the behaviour; automatic motivation refers to our emotions and impulses or habits that are relevant to the behaviour and may be activated by visual cues [23, 24]. A detailed explanation of the COM-B can be accessed online (http://www.behaviourchangewheel.com/) and in the references suggested here [23, 24].While previous research on recycling behaviour has focused on knowledge and skills (relating to Capability) and incentives of recycling behaviour (relating to Motivation), there is a scarcity of research exploring the importance of having the *Opportunity* to recycle. Specifically, we lack insights into the extent to which physical opportunity factors such as time constraints, navigation of recycling facilities in everyday contexts, the nature/design of the plastic packaging that ought to be recycled, the complexity of the behaviour(s) required to recycle, and any support or prompts provided to encourage recycling influences decisions to recycle in everyday life. A previous study used the COM-B to explore the facilitators and barriers to using a new bin scheme in a university context, finding that physical opportunity factors such as availability of a bin at the time and place it was needed (e.g. around lunch break vicinities) had an important role in determining intentions to engage with the bin scheme [25]. However, beyond this context-specific application of the COM-B there have been no in-depth explorations of the influence of Opportunity (alongside Capability and Motivational) factors on plastic recycling behaviour in everyday contexts. Furthermore, a recent review [26] highlighted that research has mainly focused on the recycling of general waste, and not plastics specifically. The limited studies which have looked at plastic, focus on drinks bottles and plastic bags [26] or on plastic waste in our oceans [27]. With increasing availability of compostable/eco plastics, there is greater variability in the plastic products that individuals are presented with in day-to-day life, which can add to the confusion about what plastic materials are recyclable [28]. Even when recycling facilities are present, cost-effective recycling of plastics depends on the ability of consumers to distinguish between and separate recyclable plastics from non-recyclable products [29]. Given this, it is imperative that we strive to understand the determinants of plastics recycling behaviour in everyday contexts.

This study aims to explore the Capability, Opportunity and Motivation factors which may determine rates of recycling of plastics behaviour from the perspectives of consumers. This will contribute to the identification of behaviour change strategies to increase recycling of plastic packaging.

## Methods

### Study design

This research reports mixed methods findings from two associated studies.

The (primary) qualitative study adopted a semi-structured interview design which was deemed most appropriate to address the overarching research question [30]. The strength of using semi-structured techniques is that it allows the researcher to gain insights to thought processes around plastic disposal, in the environmental context where the recycling is mostly taking place (e.g., in the home). This semi-directed, flexible approach allows participants to influence the direction and content of the conversation, while enabling the researcher to

explore and probe decision making processes, and degree to which associated psychological and situational factors interact. The qualitative aspects of this study have been written in line with the Consolidated Criteria for Reporting Qualitative research (COREQ) guidelines to support the sound reporting of methods and findings [31].

An online descriptive survey (secondary study) of residents living in Northern Ireland designed to examine perceptions about plastic use, and plastic food packaging was also conducted around the same time that the interviews were undertaken. This online survey was not originally intended to be used in conjunction with the qualitative findings, however the survey findings provided a useful reference point for the interview data and themes generated *post-hoc*. This survey data related particularly to opportunity and motivation components of the COM-B model and can be found later in the paper, in the results section (see Figs 2 and 3).

## Sample and recruitment

Recruitment for the qualitative interviews and the online survey took place during Autumn 2019, using convenience sampling via university websites, social media (Twitter, Facebook etc.) and word of mouth. Local supermarkets and businesses were also asked to place the flyers in their shop windows. Interested participants made contact through the researcher's university email address. For the survey, participants were able to review the information sheet and provide consent via a link to the online (Qualtrics) survey, after which they were directed to a series of questions to complete. For the interviews, potential participants were provided with an information sheet and consent form in advance and advised that interviews would be recorded. The interviews took place on university premises. Participants were reimbursed reasonable travel expenses. Before the interviews began, the participants were given the information sheet and offered the chance to ask questions. The participants were also advised that they could stop at any time during the interview. If they still wished to proceed, they completed the consent form. All of the participants lived in Northern Ireland, and lived with at least one other person, either in shared student accommodation, or with a partner. Recruitment was not limited to a specific region and thus the sample is from a broad catchment area. Age, sex and occupation of participants who took part in the interviews are displayed in Table 1. Brief demographics of survey respondents are provided in Table 2.

## Data collection

**Interviews.** Semi structured questions and the interview schedule were designed based on COM-B theoretical domains. The semi-structured interview schedule included questions and prompts of example scenarios to elicit current behaviours, beliefs, attitudes, and feelings towards a range of plastic products across familiar, everyday contexts. Some of the questions are as follows: "*What comes into your mind when you think of plastic and what we use it for; What is the relevance of recycling in your life—is it a priority for you at all, and, if not why not*?; *Ease with which they you separate waste, frequency, any barriers etc.*". The full set of questions in the interview schedule are available in S1 File.

**Table 1. Demographic breakdown of interviewees.**

|  | SEX | | OCCUPATION | Co-habiting (in a relationship or shared accommodation). |
|---|---|---|---|---|
| AGE | Male | Female | | |
| 18–24 | 1 | 4 | University UG Students | Yes |
| 25–34 | 3 | 5 | PhD students and Research Associates | Yes |
| 35–44 | 2 | 1 | University employees | Yes |
| 45+ | | 2 | University employee and One Retired member of public. | Yes |

**Table 2. Brief demographics of survey respondents.**

| AGE *(MEAN/SD)* | SEX *(FREQUENCY)* | OCCUPATION *(FREQUENCY)* |
|---|---|---|
| 40.18 / 15.60 | Male (276) | Student (217) |
| | Female (476) | Other indoor (largely desk based) work (194) |
| | | Lecturer / researcher (125) |
| | | Retired (63) |
| | | Administration (42) |
| | | Healthcare (45) |
| | | Engineer (38) |
| | | Environmental health (15) |
| | | Other outdoor work (11) |
| | | Hospitality (5) |
| | | Unemployed (1) |

Lead author and lead analyst D.Roy (DR) conducted the interviews and is an experienced qualitative researcher with a phenomenological orientation, but who also integrates this naturalistic enquiry with a realist stance [30]. This is because of her experience and knowledge of applying socio-cognitive models to explain sustainable behaviour and attitudes towards the environment. DR conducted the analysis with the support of authors M.Dempster (MD) and E.Berry (EB), by holding regular coding, cross-checking, and theme development meetings. Pre-existing assumptions were acknowledged as potentially influencing interpretation of findings due to the subjective nature of this methodology, however, biased interpretation was mitigated by members checking of codes and regular team meetings to discuss the analysis [32, 33]. The interviews lasted 45 minutes on average, and these were audio recorded. Three members of staff from the department participated, but otherwise, the participants were unknown to DR before the interviews took place.

**Online survey.**    The online survey contained a series of closed questions developed by the research team. The survey included a series of items measuring attitudes to recycling plastics, levels of concern about recycling, the effect of waste on the environment, and about who should take responsibility for reducing plastic waste. Survey items used Likert scales or yes/no responses and included questions exploring perceptions about plastic use and plastic food packaging.

## Ethics

Ethics approval for the interviews and survey was obtained through Queen's University Belfast prior to recruitment and written/typed informed consent was obtained from participants using electronic and physical consent forms.

## Data analysis

**Interviews.**    The analysis was based on a semi-directed content analysis approach [34] along with a search for emerging themes. The researchers therefore allowed for the possibility of novel patterns to emerge while also considering the findings within the context of the COM-B framework, and other relevant theoretical frameworks. This also preserved the flexibility to offer interpretations of all the data, and to allow themes to be derived from the data. DR initially familiarised herself with the data and used NVivo 12 software to assist with coding by category and consistency, and then looked for any patterns that presented. This was an

iterative process. A reflexive record was kept of the decision-making of the researcher as she coded and searched for patterns in the data.

During the readings and initial coding of the transcripts, the lead researcher allocated pieces of the narratives to nodes or categories that are relevant to the COM-B behaviour changes framework. For example, if someone described being confused by the variety of plastics that are used in food packaging, this piece of text would be coded to the broad category of capability-psychological (the coding framework is available on request from corresponding author). Further analysis of the results that were presented in the coding framework, resulted in an interpretation of meanings, which were discussed among the team on a number of occasions and final themes were agreed that captured these overall meanings for each main category. This process ensured that reflexivity was maintained throughout the analysis to support credibility and confirmability, particularly with recognition of the potential influence of researcher bias [35].

A subset of the transcripts was coded by other members of the team (MD and EB) to ensure reliability and to ensure that findings were trustworthy and practically sound. The research team (DR, MD, EB) concluded, after 18 interviews, that no new themes of note were emerging and consequently were satisfied that no further information was needed, or saturation point had been reached [34]. Moreover, data saturation was achieved when thematic categories and relationships between categories were comprehensively interrogated and interpreted. Overarching themes were finally identified and checked by the research team to confirm they provided a good representation of the findings [36]. Supplementary data can be found in S2 File.

**Online survey.** Survey responses were analysed via descriptive and frequency analysis using SPSS.

## Results

### Interview findings

A total of 18 participants were interviewed (see Table 1 for demographics). The themes extracted from the data are set within the context of the capabilities, opportunities, and motivations system (the 'COM-B' system [24, 25]. Additional supporting quotes are provided in S2 File. However, naturally there was a certain degree of conceptual overlap in the topics that participants raised. The overarching themes, framed by the COM-B constructs, were as follows: 1) Capability (psychological): 1a) 'Confusion about what plastic can be recycled'; 1b) 'Lack of information-seeking behaviour to reduce uncertainty'. 2) Opportunity: 2a) 'The need to improve labelling on food packaging'; 2b) 'Visual cues in current designs add to confusion'; 2c) 'Reducing perceived physical obstacles encourages recycling'; 2d) 'Expectancy that others should take action'. 3) Motivation: 3a) 'Activating personal motivation to recycle by increasing the visibility of the plastic waste problem'; 3b) 'Witnessing pollution damaging wildlife is an emotional nudge to reducing plastic waste'; 3c) 'There are limits to what individuals are willing to sacrifice, even among the eco-conscious'; 3d) 'Community level engagement can endorse recycling behaviours'; 3e) 'Acting on impulse'. The content coding frequency analysis can be viewed in Fig 1.

### Capability-psychological

**Confusion about what plastic can be recycled.** According to Passafaro & Stefano (2017) [37], people must be able to recognise the recyclable materials, which bin they should be placed in, and know where and how to find additional guidance when needed. A considerable barrier to recycling identified among participants, was confusion about what can be recycled, as an inaccurate understanding emerged about plastic materials themselves. This confusion was

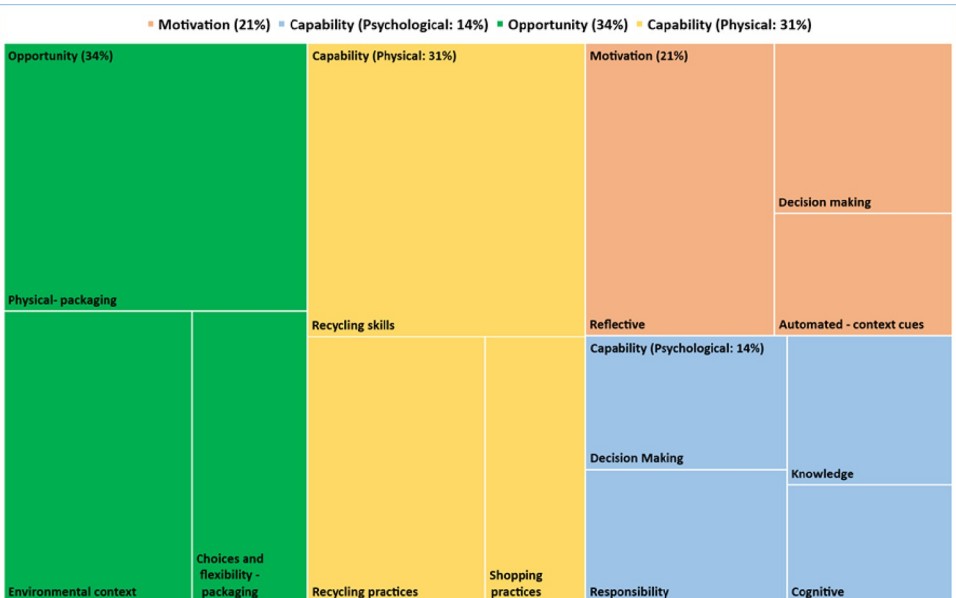

**Fig 1. A breakdown of the coding content relating to each domain, using Nvivo—according to COM-B framework.**

compounded when participants were confronted with packaging that combined several types of plastic, or other materials such as paper.

> "*I would be quite confident that I can recycle a yogurt pot or a tin or like a can from like a tin of beans and that type of thing or like beer bottles, the things that get me, are when you've got combinations of materials, like something with a film lid, like a plastic container with a film lid*". ID3.

> "*I would say, and obviously there is so much plastic that food comes in, that you're not sure, and what I would, what I tend to do, is to put it all in the recycling bin and kind of think oh well, if it can't be recycled, then it's somebody else's problem—which is maybe not the best attitude to have*" ID1

> "*Yeah, so soft plastics I would be confused, about whether you can or can't cycle, and my take was that you can't and things like your punnets for fruit, and yoghurt pots, I would be unsure about, and they would go in the black bin*" ID8.

**Lack of information-seeking behaviour, to reduce uncertainty.** Vining and Ebreo (1990) [38] found that knowledge is a significant predictor of recycling behaviour. But rather than seeking information from local authorities and recycling companies to educate themselves and aid decision making, our respondents instead appeared to routinely rely upon heuristics and routine.

> "*I have to infer, I have to check locally for recycling information because it doesn't actually say, but it's more of like a heuristic—when tired, I would never impose upon myself to go out of my way to find out more information about recycling or go to more effort. When I have individual resources depleted, like, I am tired, or things are not going well for me, if it is not made easy for me, I will just not bother*". ID23.

"*I think the limit of seeking out information is like reading the side of the packaging. 'Because, like, it's that thing where normally if you are, say cooking, you're just throwing something in the bin—it's like, second nature*". ID3

## Opportunity

**The need to improve labelling on food packaging.** Participants were asked specifically about the design of food packaging and the extent to which that helped or hindered recycling. Simple words written on the side of packaging are preferred by many; words that are clear, easy to locate and stand out, indicating whether the packaging can be recycled or not. Phrases such as "check locally" without any further details, were not considered to be helpful.

"*If there is a label that says "this is recyclable", and that is the most obvious thing because if we buy it, we obviously look at the things, and so if there is a big sticker that says recyclable then that would be good*". ID19

"*It's not clear at the minute and there are a lot of symbols and crazy instructions and things, that people feel ok about buying this because it can be recycled but the fact is that it can't easily be recycled, so I think there needs to be an awful lot more clarity in the market on some packaging*". ID26

The use of specific salient colours was also quite a popular suggestion for signalling that a package can be recycled, and this would make recycling decisions easier.

"*I do think that people on the whole are fairly visual creatures and if you can make it easily visual for them so that you don't have to think about something because obviously glass and plastic, they all have different textures to them, they feel different so people know the difference between those two. But visually though it would make it just that little bit easier than that and if it was standardized, it would be even better and people could latch on to that and symbolically of course if you go with the green, it would attract a certain number of people. . . Green obviously has been marketed very well in the last 40 years at this point, and so it has been well established if you see something that is greenish, it has that immediate connotation, whether it's true or not, of being environmentally friendly, good for the earth, "natural"*". ID10

"*The government has done things like, kids' food you've got like a traffic light kind of thing, you know, where it says, you know, green, you can eat as much of this as you want. . . and it's almost like there needs to be something as straightforward as that, that says, you know, this is fine, you can buy these things and it can be like plant based plastics, or biodegradable plastics and then these are things you shouldn't really buy unless it's necessary, as it will just, and if you must buy or have to buy you should chuck into recycling if you can, but I think there is just not enough that level of clarity around all of that*". ID12

"*I think that, you know, you have a lot of products already that can be recycled, but I think the sign on them really isn't that salient at all, in fact it is really easy to ignore it, so I think something visual in terms of colour, you know almost in the same way that food packaging now has to have a breakdown of you know, what percentage of the product contains salt, err, that kind of colour coded thing*". ID1

**'Visual cues in current designs add to confusion'.** The triangle symbol in general causes confusion, only two respondents understood its function.

"*It is always this small little thing in the corner of the bottle, that nobody really sees. It's usually like, this transparent thing on the bottom, it's usually this one, it says things like numbers like 1,3,5 or 7, it's not really telling you whether it's recyclable or not. I don't see any reason for people putting that on, because people are not going to pay attention to it*". ID19

"*. . .but I do think the two arrows that are on it at the minute, the chunky arrows, I take that as this can definitely be recycled, but then on fruit punnets there is that kind of triangle of arrows which I take to mean it can be recycled in some areas but I don't know if that is right, and I would generally think, it's probably unlikely that this is recyclable in my area, so I feel that specific kind of triangle of arrows doesn't work, it's too ambiguous*". ID8

Recycling bins are provided in colours to indicate to consumers what materials should be placed in them but there is no standardisation of bins across the country. Participants highlighted the fact that different coloured bins are used by different local authorities (Councils) and that different materials can be recycled in these different bins. They reported annoyance with this changing system and that it would be easier for householders if the recycling system was standardised.

"*It varies from Council area to Council area which I think. . ., is just crazy, why there can't be just a unified approach*". ID21

"*I think it's really annoying that in some places you can recycle your glass, and some places you can't, and I think we would catch more materials in this segregated [referring to stackable recycling boxes]*". ID24

Some participants suggested that, in the absence of a standardised system, there should at least be clear information about which bin was used for which materials and that this information should be obvious at the point of recycling, rather than requiring the person to find the information on a website.

"*If the information isn't presented to me at the forefront, when I need it, I just throw that into the black bin, the black bin being the general waste bin*". ID13

"*So putting big stickers on the bins that really tell you what you are meant to recycle and give us even more bins, for more different types of plastics*". ID25

"*Whenever I get home, I've been working all day, I've got the dinner to put on, got the kids to get sorted, home work to do, I really don't want to be standing at my recycling bin going, can this go in, I know that can't go in, you want it to be as relatively straightforward as you can, otherwise it's just becomes another complication in your life, and you are already working with time constraints*". ID21

**Reducing perceived physical obstacles encourages recycling.**   Some participants identified physical barriers to recycling that resulted in materials not being recycled appropriately. The obstacles identified by participants suggest that any inconvenience could be a demotivating factor that will influence behaviour.

"*Yes we do [have a food caddy], and it was used for a while, until we ran out of the little bin liners, because you have to get the special size, or the ones that decompose, and they haven't been bought since. If there were the little bin liners we would still be using that caddy, for sure.*" ID8

"*We had missed one of the bin days, so we had some of those recyclable bin bags and we put some of our recycled stuff in and set them out but they just left it there because it wasn't in the boxes but we didn't realise so we had to go to the recycling somewhere and that was a bit annoying, but like, for four people in a house, one of those fills pretty quickly*". ID20

**Expectancy that others should take action.**    Third parties (government and business) were blamed for not doing more to encourage recycling. This resulted in a paradox in the responses, where more government regulation and less plastic manufacturing was advocated, hence removing choice, but more choice of products provided by retailers was also supported. The unifying element of this paradox is that participants were expecting action by perceived powerful others that would cause a change in their behaviour.

Manufacturers of plastic products were perceived to have some responsibility for any problems caused by plastic pollution. Participants suggested that manufacturers should produce less plastic or find alternative forms of plastic, and, in some cases, this was seen to be more effective than individual behaviour to deal with plastic pollution.

"*I think we should be going along the lines of encouraging people to come up with the better packaging product rather than coming up with a solution at the end to what do we do with it*". ID18

"*I just find it frustrating because I just think it's unnecessary and I think that a) why does this stuff have to end up in the sea, or at the side of the road, so there's that, and then b) why is some of this stuff actually still being made and purchased in these quantities when it's perfectly possible to get away, perfectly possible to live without producing some of the stuff in the first place so yeah, I just find it, it's a frustrating situation but one which seems to be getting kind of worse probably, than better I think*". ID12

Local and central government were seen to have several roles in managing plastic consumption and recycling. These were: Educating citizens about recycling; rewarding people to motivate recycling; introducing a levy or ban on plastic products to discourage plastic use or encourage re-use, and influencing retailers' use of plastics.

"*I still think not enough is done by individual council's to make people aware of the importance of recycling, and definitely not enough is done to make people aware of how to recycle properly, and it's something that is obviously is in all the council's interests that they really should be putting more money into this because if you don't change the way people think about recycling how are you going to get people more likely to recycle*". ID1

"*It would be good for central government to legislate for the big supermarkets not to be using, to cut down on the amount of plastic packaging that they use.*" ID13

Participants suggested that retailers could assist the recycling effort by using recyclable materials more frequently, providing bins for recyclable materials and not encouraging people to buy plastic by offering cheap plastic products.

"*But it also comes back to fast food or takeaway, or sandwich places, I think all of their products should be based on recyclable and there should be one bin once you have finished eating whether it's a container or a fork, a wrap, it should all be put in the one bin, and the bin should be a compostable bag where it's all taken away*". ID18

"*I think they do need to make some changes in the law, you know [retailer] their coffee cups, I have my own reusable cup anyway, but I noticed they're using Veg-ware so they are compostable, and I think will it's possible to do that, so why can't other chains do that as well*". ID26.

## Motivation

**Activating personal motivation to recycle by increasing the visibility of the plastic waste problem.**   The impact of visual information was identified as a powerful way of encouraging people to engage more in recycling behaviour. Indeed, the absence of visual indicators of the consequences of not recycling was suggested as a reason why people do not engage in recycling to a greater extent.

"*If that was right next to the entrance to your housing estate there was this huge, big landfill, just like stinking and sitting there, like everywhere you know what I mean. We just put it all to the back of our minds and I think we do that with everything, like we ignore things that aren't right in front of us, and like it's just like that kind of chosen ignorance I think is a big part of it*". ID3

"*The visual aspect really works in that case—it does work for me. We can go back to that quote that "one person's death is a tragedy, but that a million is just a statistic". It's also why I think film and any kind of creative art is, when it is effective, is effective because it takes the personal and makes it universal*". ID10

**Witnessing pollution damaging wildlife is an emotional nudge to reducing plastic waste.**   Emotions evoked from seeing wildlife being hurt was a powerful motivator of plastic recycling behaviour. Participants could provide examples of how they had been motivated to engage in recycling behaviour because of their emotional responses to observing how plastic pollution has damaged non-human animal life.

"*The animal part touches me, because I'm a vegetarian and I eat a mostly vegan, and it's definitely very sad to see the animals being harmed. . . I didn't care if it was very hard to see they were showing things that were really gruesome, and that was the turning point for a lot of my friends and I, because you're finally seeing it, it wasn't just plastic on a beach, which is sad, but you're seeing something else suffering because of it, and you feel the impact more, because I mean any kind of living thing you don't want to hurt them, well, most people don't want to hurt them, but definitely the animals, that hits home more especially for me*". ID22

"*I actually read an article about the typhoon of plastic that is in the oceans at the moment, and how many issues this causes, particularly as well with the highlighted damage of plastic straws, for sea life all these stories together have sort of really highlighted plastic as a negative environmental thing and that is why when I'm thinking about plastic, the first thing I normally think about is all these stories about how animals are being harmed by the production of and wasting of plastic or I should say the incorrect disposal of plastic*". ID23

**There are limits to what individuals are willing to sacrifice, even among the eco-conscious.**   In general, participants reported a desire to do the right thing and emphasised the importance of recycling plastics. Indeed, some participants indicated that a clear visual cue such as a label indicating that a product was recyclable would make it more likely that they would purchase that product:

*"So I have changed products recently to things that specifically market themselves as being completely recyclable, made of recycled products, very low packaging, and they have literally written that, all over it. It was very, very obvious and that completely made me swap my choices". ID8*

*"I was buying some coffee the other day, beans, coffee beans and it was in a kind of, you know, one of those foil kind of bags that coffee beans come in, and then it had a paper kind of surround which is just a branding thing and I just felt, I just thought, that is just completely. . . so I didn't buy that, so you kind of start to make some choices about things you cannot buy, and you think actually, that's just pathetic, so you don't buy it". ID12*

However, the effort needed to reduce plastic use was not without limitations, and an important limitation was cost.

*"If there was a choice, if there was, you can get milk or orange squash or something in a bottle that's biodegradable or one that isn't, then I'd definitely choose the biodegradable one, even if it meant, if it was a little bit more, not if it's like, a lot more, but a little bit more, or something like that ". ID12*

*"But now I would probably spend a little bit more, but if it was, the difference of a £1.40 per bottle versus £1 per bottle, I don't think I could get over that, even though it's 40p it's 40% of the price of the alternative". ID23*

**Community level engagement can endorse recycling behaviours.** In addition, some participants also identified the important role of communities in influencing behaviour. Here, the emphasis was on social modelling rather than imposition.

*"I have got a lot of friends who are quite ecologically conscious and to a large extent are vegans for ecological reasons or have been lifelong vegetarians and don't own cars and things for ecological reasons. . . so that's the sort of notion of being on the periphery of this community and knowing about its existence, it's sort of it's nice in general to know that there is a collection of people with shared, well, all collections of people or individuals with shared values that form a group but there's some group formation now, I think more so than ever around ecological and green sustainability issues". ID23*

*"If community leaders were trying to do something, I think, if these problems were framed as a problem for our community, then, and there might be ways of, sort of massaging the problem such that you talk about impact on the local community. The trouble is of course that it's not on the local community, not at all, there is a wider impact, but there might be a way of appealing to people's civic pride at a local community level and dealing with things that way for instance". ID4.*

**Acting on impulse.** Many of our routine household activities such as shopping and disposing of waste are performed without much thought [39]. Consequently, breaking old habits such as being seduced by attractive containers while shopping instead of trying to avoid plastic packaging, is not easy to stop without some conscious effort.

*"And also being environmentally friendly too, they see, for example my friend said don't do that because dot, dot, dot and all the information was there, but all the information was there, but I was not conscious, I was not awake but she told me this, this, and this and she showed me how to do it. So then I was ok, this is right, ok, I am going to change that" ID16.*

*"If strawberries come in, like a very light punnet, or if they're all wrapped up and made to look very nice and presentable, I think I would still go for the very nice and presentable ones, I would give up my ethics—unless I am consciously thinking about it of course"* ID1.

A further limitation was that participants reported acting on impulse when shopping. In other words, not taking the time to determine which products are packaged in recyclable material, but buying products out of habit, or because the packaging makes them convenient, or whatever product is most accessible, given that they are often shopping under time constraints.

*"While in the supermarket, the different levels of decisions whenever I'm in the supermarket I wouldn't ever think that deeply about it, I don't think I would make decisions primarily on the anticipation of the eating and preparing the food and not acknowledging packaging".* ID23

*"In terms of packaging, the kind of things I would buy have quite a short shelf life, and so I wouldn't necessarily pay a lot of attention to packaging. . .I think in terms of purchasing food, I would say it is about 95% about the actual product, and probably about 5% thinking about the packaging. I think maybe a few years ago, I probably did pay more attention.".* ID1

## Survey findings

Although the online survey was not originally intended to be used in conjunction with the qualitative findings, following analysis of the interview data, we identified relevant questions from the online survey that helped to corroborate the qualitative findings. There were four questions in the survey that are particularly relevant to this study, which examined opportunity and motivation as possible determinants of recycling. Responses were received from N = 756 participants and the key findings are presented in Figs 2 and 3. The frequencies observed

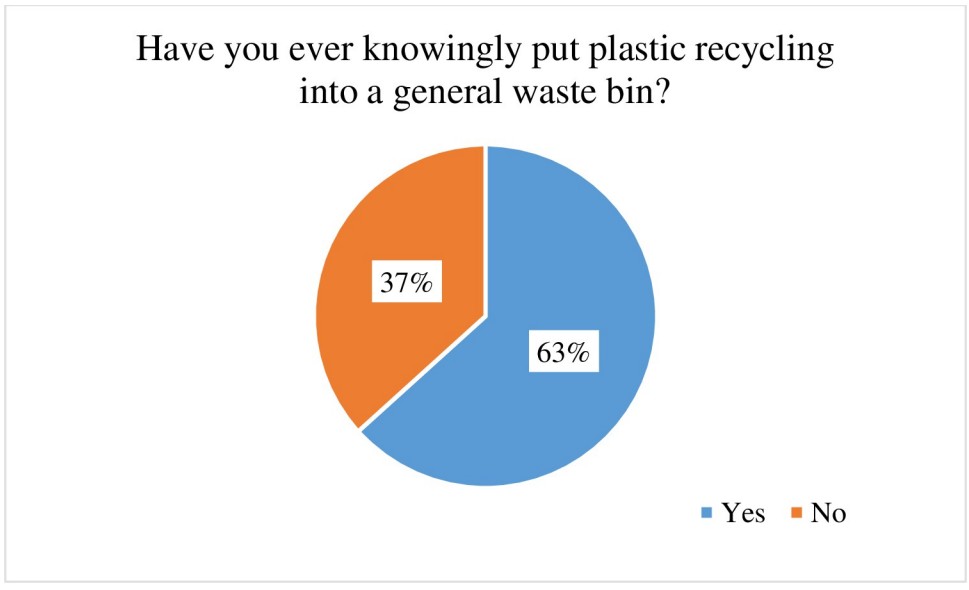

**Fig 2. Responses to online survey selected items (motivation and opportunity) (agree and strongly agree percentage scores are combined).**

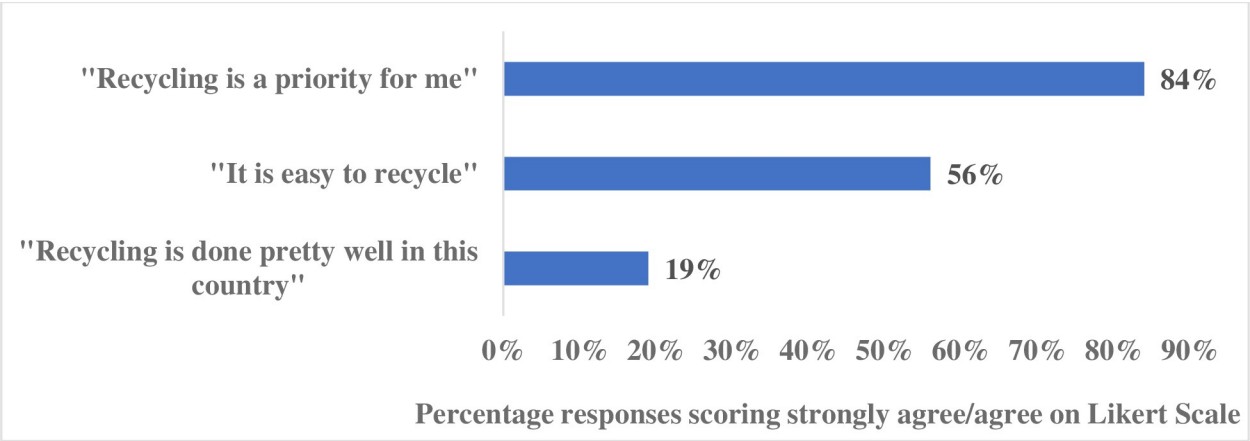

**Fig 3. Responses to online survey selected items (recycling behaviour).**

reinforce importance of themes extracted relating to the reduction of physical obstacles to increase accessibility and ease of recycling in practice, and the need to motivate recycling in creative ways. One of the questions asked if they had ever knowingly put plastic recycling into a general waste bin and 63% admitted that they had consciously placed recyclable items in the general waste bin (See Fig 2). And yet 84% of respondents answered that recycling was a priority for them, with 56% saying they found it easy to recycle (see Fig 3). Only 1 in 5 survey respondents felt that recycling was carried out to a "good" standard, suggesting that most felt pessimistic toward recycling capabilities in Northern Ireland (See Fig 3).

## Discussion

This study sought to examine the facilitators and barriers to plastics recycling by consumers, using the COM-B model as a guide. The findings of the content analysis reveal that almost two thirds of the data fall under the headings of physical capability and physical opportunity, with psychological capability and reflective motivation accounting for remaining third of the data. There is less evidence in the data about the role of automatic motivation or social opportunity. Overall, the main themes point to a knowledge gap; a lack of understanding and confusion about how to recycle plastic waste properly (capability) and unclear recycling instructions (capability and opportunity). This co-exists with a lack of willingness to seek out such instruction that is provided by local authorities because recycling is often not perceived to be a priority in the wider context of busy day to day lifestyles and responsibilities (motivation). Witnessing plastic pollution harming marine life and places of natural beauty, for some, is an emotional nudge to reducing plastic waste and could activate personal motivation to recycle. But without these visual cues, our respondents lack sufficient motivation to seek the detailed information needed about local recycling policies and educational materials (motivation). The recycling of plastic waste is a complex behaviour and poses specific challenges for householders (capability and opportunity). The reality is many simply do not want to be faced with uncertainty and confusion when trying to perform what, to them, should be made an easy, effortless behaviour.

Participants in the study indicated that the current labelling on plastic packaging adds to that confusion. The confusion expressed by participants has been found in previous research. WRAP (2020) [40] found that the most frequently cited barrier to recycling in the UK is 'uncertainty about what can/can't be recycled'. Information/knowledge about what can be

recycled has also been identified as one of the four main barriers to recycling in a systematic review [41]. In South Africa, lack of knowledge was the second most cited barrier to recycling, behind lack of time [42]. This latter barrier highlights the importance of information being readily accessible (i.e. clear labelling on the plastic packaging itself). Providing information about what can be recycled elsewhere (not on packaging) is unlikely to be helpful as people are very reluctant to take the time to familiarise themselves with this information. This information needs to be accessible at the point of plastic recycling. People are less likely to be confused about whether a plastic package can be recycled if they immediately recognise recycling advice when they look at the package [43]. Being faced with uncertainty for some does not result in information seeking behaviour, rather it causes frustration, resulting in the plastic product ending up in general waste. This adds weight to the importance of visual, unambiguous cues about the recycling of plastic packaging to remove barriers in the form of frustration.

Not surprisingly, then, one of the solutions offered by participants, to improve recycling of plastics is to have clearer labels on the packaging. Participants reported that existing labels are unhelpful and can only add to confusion. They prefer clear, unambiguous labels and colour coding was also suggested. Labelling about recycling on packaging has previously been found to be a cause of confusion and a determinant of successful recycling [44].

Colour coding of bins was also suggested as a way of helping to identify which types of recycling waste is placed, in which bin. The colour of bins and other aspects of recycling bin design are associated with appropriate separation of waste and acceptability of a recycling scheme [15, 45]. This issue is considered important enough to involve the public in the design of recycling bins in Greece [46]. Participants in the present study additionally indicated that it would be preferable for the colour of bins and what should be placed in that bin to be standardised throughout the country. Again, this is an example of how making things easy at the point of recycling, so that waste separation becomes a habitual process, is desirable. On this basis, Burgess et al., (2021) [47] have recommended using one bin for all household plastics recycling, although they acknowledged that this will impact on other parts of the recycling process.

Although the participants in the present study were positive about recycling, this turns to negativity when recycling is not easy. In other words, there are limits to the sacrifices or efforts that individuals are prepared to make to recycle plastics. Recycling is also less likely to happen as financial or time costs increase. Our data does not allow us to estimate the strength of this relationship or to determine whether the relationship is linear, but the data suggests that the relationship exists, in some form. Our data also points to the need for a further in-depth investigation of the decision-making processes involved around plastic waste disposal.

Our study signals that participants expected government, manufacturers and retailers to do more to manage the amount of plastic in society. This indicates a belief that addressing the reduction and recycling of plastics is not within the locus of control of the individual. An internal locus of control has been shown to be related to environmentally responsible behaviour [48], so these findings might give some cause for concern. However, more recent research suggests that both internal and external control are associated with pro-environmental behaviour [49]. Therefore, perhaps participants are simply expressing a credible view that everyone involved in the circular plastics economy must play a role in improving the situation.

While successful plastic recycling largely depends on public behaviour, retailers can have a powerful impact on consumers purchasing environmentally conscious products [50], and government levies can be effective in reducing plastic waste [51], although government intervention is not by itself a panacea [52].

What this research highlights, are that contextual factors, such as waste disposal infrastructure, and the characteristics of food packaging, are having a significant negative impact on plastic recycling behaviour. Obvious important action, rather than relying upon websites to

convey recycling information to householders and consumers by third parties, is to also model the importance of plastics recycling [53]. So, if third parties (manufacturers, government, retailers) take action to make recycling easier, this could have a twofold impact on consumers' behaviour.

This research focuses on exploring barriers and facilitators to recycling plastics, however it is worth acknowledging the reality that recycling processes involved do not necessarily offer a clean solution to the issue of excessive use of plastics. Virtually all plastic packaging contains additives which are chemicals that help to preserve the package and support longevity [54]. However, these additives can be emitted into the air, soil, water systems and our food chain if packaging is not recovered and reprocessed in a way that mitigates transmission of these additives [54]. Thus, while encouraging recycling behaviour is an important step toward reducing plastic waste disposal, it is important to consider the role of infrastructure and recycling expertise in ensuring that plastic recycling processes are environmentally safe.

## Conclusion

The results suggests that the negative impacts of not recycling are easy to set aside because there is no ongoing, visual reminder of this impact, and responsibility is not equally shared among all those who have a role to place in the value plastics chain. The recycling of plastic waste is a complex behaviour and poses specific challenges for householders who simply do not want to be faced with uncertainty and confusion when trying to perform what, to them, should be made an automatic, effortless behaviour. It seems having to make an extra effort becomes a barrier, and good intentions to recycle will be set aside. The upshot of this is that visible impact of plastics on the environment needs to feature regularly when promoting plastic waste recycling to the public, and the act itself needs to be made as convenient as possible, to maximise the behaviour.

The primary recommendations from this research, guided by the BCW [24], which will involve action by government and manufacturers and consultation with retailers and consumers are:

1. Increasing capability and opportunity to recycle: Plastic packaging should have visual, unambiguous cues about whether it can be recycled (intervention functions include 'enablement'–environmental restructuring making it easier to recycle e.g. visual cues on labels; and 'education'–clear and accessible information about what can be recycled).

2. Increasing capability and opportunity to recycle: Bins should be colour coded to assist with the separation of waste and it would be beneficial if these colours were standardised throughout the country and matched with colour coding on packaging (intervention functions include 'enablement'–environmental restructuring making it easier to recycle e.g. visual cues on bins).

3. Increasing motivation to recycle: Harnessing the power of habit and increasing the visibility of the impact of (not) recycling (intervention functions include 'enablement' (as mentioned above); and 'persuasion'–visual cues as reminders of impact of not recycling or the positive impact of recycling and reinforcement of actions.

## Supporting information

**S1 File. Interview schedule (this supporting information file contains the minimal dataset required to replicate the interview data collection).**
(DOCX)

**S2 File. Supplementary data file (this supporting information file contains the minimal dataset required for preliminary qualitative analysis).**
(DOCX)

## Acknowledgments

We thank all the participants for donating their time to help with this study.

## Author Contributions

**Conceptualization:** Emma Berry, Martin Dempster.

**Formal analysis:** Deborah Roy, Emma Berry, Martin Dempster.

**Funding acquisition:** Emma Berry, Martin Dempster.

**Investigation:** Deborah Roy.

**Methodology:** Deborah Roy, Emma Berry, Martin Dempster.

**Project administration:** Deborah Roy, Martin Dempster.

**Resources:** Martin Dempster.

**Software:** Deborah Roy.

**Supervision:** Martin Dempster.

**Validation:** Deborah Roy, Emma Berry, Martin Dempster.

**Visualization:** Deborah Roy.

**Writing – original draft:** Deborah Roy, Emma Berry, Martin Dempster.

**Writing – review & editing:** Deborah Roy, Emma Berry, Martin Dempster.

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
