## [Decision Letter · Decision Letter 0]

16 Dec 2021

PONE-D-21-36666“If it is not made easy for me, I will just not bother”. A Qualitative Exploration of the Barriers and Facilitators to Recycling PlasticsPLOS ONE

Dear Dr. Berry,

Thank you for submitting your manuscript to PLOS ONE. After careful consideration, we feel that it has merit but does not fully meet PLOS ONE’s publication criteria as it currently stands. Therefore, we invite you to submit a revised version of the manuscript that addresses the points raised during the review process.

ACADEMIC EDITOR: Please carefully consider the comments of both reviewers and address them as they are required for acceptance. Please submit your revised manuscript by Jan 30 2022 11:59PM. If you will need more time than this to complete your revisions, please reply to this message or contact the journal office at plosone@plos.org. Please include the following items when submitting your revised manuscript:A rebuttal letter that responds to each point raised by the academic editor and reviewer(s). You should upload this letter as a separate file labeled 'Response to Reviewers'.A marked-up copy of your manuscript that highlights changes made to the original version. You should upload this as a separate file labeled 'Revised Manuscript with Track Changes'.An unmarked version of your revised paper without tracked changes. You should upload this as a separate file labeled 'Manuscript'.

We look forward to receiving your revised manuscript.

Kind regards,

Reginald B. Kogbara, Ph.D.

Academic Editor

PLOS ONE

Journal Requirements:

"This work was supported by the Engineering and Physical Sciences Research Council (ESPRC) Grant number EP/S025545/1.

" ext-link-type="uri" xlink:type="simple">https://epsrc.ukri.org/"

"We thank all the participants for donating their time to help with this study. This work was supported by the ESPRC [Grant number EP/S025545/1]."

"This work was supported by the Engineering and Physical Sciences Research Council (ESPRC) Grant number EP/S025545/1.

" ext-link-type="uri" xlink:type="simple">https://epsrc.ukri.org/"

Reviewers' comments:

Reviewer's Responses to Questions

**Comments to the Author**

1. Is the manuscript technically sound, and do the data support the conclusions?

Reviewer #1: Partly

Reviewer #2: No

2. Has the statistical analysis been performed appropriately and rigorously? 

Reviewer #1: No

Reviewer #2: I Don't Know

3. Have the authors made all data underlying the findings in their manuscript fully available?

Reviewer #1: Yes

Reviewer #2: Yes

4. Is the manuscript presented in an intelligible fashion and written in standard English?

Reviewer #1: Yes

Reviewer #2: No

5. Review Comments to the Author

Reviewer #1: In my opinion, the authors should put the procedure for gathering data in one method first then separate between direct interview and by online. In this case, author mentioned total number of respondents first then separate. In this manuscript, authors seemed set aside online respondents. It was looked from their statement in page 24 (A separate survey...). From this point, authors seemed less satisfaction with number of respondents of direct interview or the results of direct interview. Authors should be neutral, so respondents (both direct interview and online) should have the same position. We have to respect all of their opinion.

Or authors can compare between direct respondents versus online. So in this case statistical analysis are needed.

Further, authors should add a proof or a study that plastics can/ can not be recycle. Here, terminology of recycle has to be explained whether just re-use, re-pair, or fully re-cycle to be different form, such as become a WPC (wood plastics composite) product. Recycle here means what?

In addition to recycle, did the authors think plastic can be degraded in nature? Issue of microplastics in the environment have been hot nowadays. Some sentences were needed to explain more, such as impact of recycling the plastics (pollution in soil and air/ because of plastics combustion), benefits or drawback of these actions, etc

Reviewer #2: The paper investigates an important issues using COM-B model. However, the introduction and method section is very confusing. A through revision of introduction and method is required before going through the result.

6. PLOS authors have the option to publish the peer review history of their article (what does this mean?). If published, this will include your full peer review and any attached files.

Reviewer #1: No

Reviewer #2: No

---

## [Author Response · Author response to Decision Letter 0]

28 Jan 2022

Dear Reviewers and Editor for PLOS ONE,

Thank you very much for taking the time to read and comment upon our paper.

Please find a detailed response to each of your comments in the tables enclosed in the document entitled 'Response to Reviewers', along with a ‘Revised Manuscript with Track Changes’ version of the new manuscript where your suggested changes have been visibly made. Also, we have provided an unmarked final version named ‘Manuscript’. Please view the ‘All mark-up’ version of the tracked changes document or refer to the 'clean manuscript' in the PDF proof and this way, the line and page numbers will align with each specific comment addressed.

We have attempted to respond to the comments you made in the PDF version of the submitted manuscript first which is Section 1 of the document 'Response to Reviewers'.

This is followed by our response to the main comments sent to us in a word document. They overlap in some instances, and so we cross referenced where this is the case.

All the changes are in the ‘Revised Manuscript with Track Changes’ paper and so any line and page numbers we provide to direct you relate to this tracked change document and not the final clean manuscript.

Furthermore, we have implemented changes as instructed by the editor: we have adhered to the PLOS ONE file and document formatting; additional information regarding the funders input is as follows “The funders had no role in study design, data collection and analysis, decision to publish, or preparation of the manuscript”; and we have removed funding information from the acknowledgments section in the manuscript.

Best wishes,

Emma

---

## [Decision Letter · Decision Letter 1]

9 Mar 2022

PONE-D-21-36666R1“If it is not made easy for me, I will just not bother”. A Qualitative Exploration of the Barriers and Facilitators to Recycling PlasticsPLOS ONE

Dear Dr. Berry,

Thank you for submitting your manuscript to PLOS ONE. After careful consideration, we feel that it has merit but does not fully meet PLOS ONE’s publication criteria as it currently stands. Therefore, we invite you to submit a revised version of the manuscript that addresses the points raised during the review process.Please consider both reviewers' comments carefully, especially regarding the methods section discussing the surveys employed, and address the issues raised as they are required for acceptance. Please submit your revised manuscript by Apr 23 2022 11:59PM. If you will need more time than this to complete your revisions, please reply to this message or contact the journal office at plosone@plos.org. Please include the following items when submitting your revised manuscript:A rebuttal letter that responds to each point raised by the academic editor and reviewer(s). You should upload this letter as a separate file labeled 'Response to Reviewers'.A marked-up copy of your manuscript that highlights changes made to the original version. You should upload this as a separate file labeled 'Revised Manuscript with Track Changes'.An unmarked version of your revised paper without tracked changes. You should upload this as a separate file labeled 'Manuscript'.If applicable, we recommend that you deposit your laboratory protocols in protocols.io to enhance the reproducibility of your results. Protocols.io assigns your protocol its own identifier (DOI) so that it can be cited independently in the future. For instructions see: https://journals.plos.org/plosone/s/submission-guidelines#loc-laboratory-protocols. Additionally, PLOS ONE offers an option for publishing peer-reviewed Lab Protocol articles, which describe protocols hosted on protocols.io. Read more information on sharing protocols at https://plos.org/protocols?utm_medium=editorial-emailutm_source=authorlettersutm_campaign=protocols.

We look forward to receiving your revised manuscript.

Kind regards,

Reginald B. Kogbara, Ph.D.

Academic Editor

PLOS ONE

Journal Requirements:

Reviewers' comments:

Reviewer's Responses to Questions

**Comments to the Author**

1. If the authors have adequately addressed your comments raised in a previous round of review and you feel that this manuscript is now acceptable for publication, you may indicate that here to bypass the “Comments to the Author” section, enter your conflict of interest statement in the “Confidential to Editor” section, and submit your "Accept" recommendation.

Reviewer #1: (No Response)

Reviewer #2: (No Response)

2. Is the manuscript technically sound, and do the data support the conclusions?

Reviewer #1: Partly

Reviewer #2: Partly

3. Has the statistical analysis been performed appropriately and rigorously? 

Reviewer #1: No

Reviewer #2: No

4. Have the authors made all data underlying the findings in their manuscript fully available?

Reviewer #1: Yes

Reviewer #2: No

5. Is the manuscript presented in an intelligible fashion and written in standard English?

Reviewer #1: Yes

Reviewer #2: No

6. Review Comments to the Author

Reviewer #1: Please response my comments both specifically and in the manuscript. This is important although your second draft of manuscript is better now. My previous comments were:

In my opinion, the authors should put the procedure for gathering data in one method first then separate between direct interview and by online. In this case, author mentioned total number of respondents first then separate. In this manuscript, authors seemed set aside online respondents. It was looked from their statement in page 24 (A separate survey...). From this point, authors seemed less satisfaction with number of respondents of direct interview or the results of direct interview. Authors should be neutral, so respondents (both direct interview and online) should have the same position. We have to respect all of their opinion.

Or authors can compare between direct respondents versus online. So in this case statistical analysis are needed.

Further, authors should add a proof or a study that plastics can/ cannot be recycle. Here, terminology of recycle has to be explained whether just re-use, re-pair, or fully re-cycle to be different form, such as become a WPC (wood plastics composite) product. Recycle here means what?

In addition to recycle, did the authors think plastic can be degraded in nature? Issue of microplastics in the environment have been hot nowadays. Some sentences were needed to explain more, such as impact of recycling the plastics (pollution in soil and air/ because of plastics combustion), benefits or drawback of these actions, etc.

Reviewer #2: I would like to thank Authors for the revision. The paper has improved. However, the manuscript needs additional work, a major revision. This includes clear explanation of methods and surveys, presenting result in more effectively and improving the readability. I have comments in PDF version as well as separate comment by section in word file.

7. PLOS authors have the option to publish the peer review history of their article (what does this mean?). If published, this will include your full peer review and any attached files.

Reviewer #1: No

Reviewer #2: No

---

## [Author Response · Author response to Decision Letter 1]

4 Apr 2022

Response to Reviewers

Thank you for the comments provided. Below we have detailed how each comment has been addressed (red text) and we have marked changes in the ‘Manuscript with Track Changes’. Page numbers below align with the ‘Simple Mark-up’/ clean version of the document.

Reviewer 1:

The second draft of manuscript is better now, unfortunately I cannot find the specific answers or responses of the authors to my previous comments as copy paste below:

In my opinion, the authors should put the procedure for gathering data in one method first then separate between direct interview and by online. In this case, author mentioned total number of respondents first then separate. In this manuscript, authors seemed set aside online respondents. It was looked from their statement in page 24 (A separate survey...). From this point, authors seemed less satisfaction with number of respondents of direct interview or the results of direct interview. Authors should be neutral, so respondents (both direct interview and online) should have the same position. We have to respect all of their opinion.

Or authors can compare between direct respondents versus online. So in this case statistical analysis are needed. 

Thank you for these comments – we recognise the ambiguity of the survey methods and results and with respect to how these fit within the larger report and relate to the qualitative findings. We have separated the methods for each study and provided greater detail on each (see changes on pages 6-11) as well as more detail on the survey data results. Furthermore, we have separated and reworded the description of the survey findings (see pages 25/26).

Further, authors should add a proof or a study that plastics can/ cannot be recycle. Here, terminology of recycle has to be explained whether just re-use, re-pair, or fully re-cycle to be different form, such as become a WPC (wood plastics composite) product. Recycle here means what?

Thank you for this comment – we recognise that the term recycle carries with it certain assumptions and therefore we have added a definition in the introduction for the purposes of this research, which aligns with how we define and use this term with respect to the findings presented (see page 3/4 which we have also pasted here): “For clarity, for the purpose of this research we define (plastic) recycling behaviour as the act of disposing of a used (or no-longer-being-used) plastic packaging in a bin or other resource which is intended to recover that packaging for reprocessing.”

In addition to recycle, did the authors think plastic can be degraded in nature? Issue of microplastics in the environment have been hot nowadays. Some sentences were needed to explain more, such as impact of recycling the plastics (pollution in soil and air/ because of plastics combustion), benefits or drawback of these actions, etc.

Thank you for this comment – we recognise that the impact of recycling itself can be harmful and we have added detail on this in the ‘discussion of results’ section to propose additional considerations in the balance of the discussion (see page 30). We have also added a sentence in the introduction, which takes into account other preventative behaviours including the reuse and reduction of use of plastics (see page 3).

Please response these comments first. Thank you.

Reviewer 2:

I would like to thank Authors for the revision. The paper has improved. However, the manuscript needs additional work, a major revision. This includes clear explanation of methods and surveys, presenting result in more effectively and improving the readability. I have comments in PDF version as well as separate comment by section in word file.

Thank you for taking the time to review our manuscript for a second time. Firstly, we have addressed he comments provided in the manuscript document, some of which overlap with the changes made to the points below.

1. Abstract:

The author has revised the abstract. However, the revision is not sufficient. Currently abstract is long. Abstract need further work by directly focusing on the aim of the study, method and the key findings. 

Thank you for this comment – we have revised the abstract in line with the changes suggested.

2. Introduction:

The introduction has improved but can benefit from one more round of revision especially removing long and complex sentences with the short and meaningful sentences and removing information that doesn’t add the discussion. 

We have removed or shortened long and redundant sentences throughout the introduction (however we have now also moved the description of the COM-B model from the methods to the introduction as we feel it is better placed earlier to provide context for the discussion of the model).

3. Method

• The first paragraph, Is this paragraph should be revised to explain the strength of qualitative methods in the COM-B framework? and its advantage for this study using peer-reviewed sources. [This comment is also in the pdf version]. We have revised the start of the methods section in line with this suggestion – as mentioned, we have also the description of the COM-B model from the methods to the introduction as we feel it is better placed earlier to provide context for the discussion of the model (see changes on pages 6).

• The author should sufficiently explain Why a semi-structured interview is deemed most appropriate for the study. 

This is described on page 6.

• The method section should discuss both the surveys: online and interview and justify why two separate data collection process is implemented and how these two address the research question 

We have now separated the methods for each study and have provided greater detail on each. We also provide a rationale for including both studies in this paper with respect to the research aims (see changes on pages 6-11). 

4. Result and discussion

a. Introducing a new online survey in the middle of the result leaves the reader in confusion. Author should discuss all data collection strategies and the reasons for using two data collection strategy in the method section. 

As mentioned, we have now separated the methods for each study and provided greater detail on each, to include the rationale for including both studies in this paper (see changes on pages 6-11). 

b. Also, the result of the two surveys can be presented in different subsections. 

We have separated and reworded the description of the survey findings in the results section (see pages 25/26).

c. As mentioned, the online survey has 756 respondents. The survey can provide meaningful insight into the study. However, the result is not well presented. 

As above, we have added to the description of the survey findings (see pages 25/26).

d. I would suggest the following: Insert a paragraph discussing the survey process (if it is secondary information) mentioning the source of information is helpful. Also, use information in Table 2 like age and sex to discuss the participant characteristics. Then in the result insert an additional subsection and present the result. A chart may be easy to understand. 

Thank you for this suggestion, we have highlighted that this study is secondary on pages 6/7, and as mentioned above, we have separated the methods for each study and provided greater detail on each. Furthermore we have added a table displaying brief demographics for survey participants along with a short description of this (see changes on pages 6-11).

e. Also, if there is a separate discussion, the section heading should result only not result and discussion. So doing, the author must shift any discussion from the result section to the discussion section. 

We have removed the term discussion from the results heading (page 11).

f. Table 2 need revision. Please use a separate table to provide information on a variable with a different structure. For example, responses collected as Likert scale and yes/no(binary) can be separated and presented in a graph chart or different table. 

We have divided the questions that use different response structures into separate tables (see changes on pages 26).

---

## [Editor Report · Decision Letter 2]

6 Apr 2022

“If it is not made easy for me, I will just not bother”. A Qualitative Exploration of the Barriers and Facilitators to Recycling Plastics

PONE-D-21-36666R2

Dear Dr. Berry,

We’re pleased to inform you that your manuscript has been judged scientifically suitable for publication and will be formally accepted for publication once it meets all outstanding technical requirements.

Kind regards,

Reginald B. Kogbara, Ph.D.

Academic Editor

PLOS ONE
---

## [Editor Report · Acceptance letter]

13 Apr 2022

PONE-D-21-36666R2 

“If it is not made easy for me, I will just not bother”. A Qualitative Exploration of the Barriers and Facilitators to Recycling Plastics 

Dear Dr. Berry:

I'm pleased to inform you that your manuscript has been deemed suitable for publication in PLOS ONE. Congratulations! Your manuscript is now with our production department. 

Kind regards, 

on behalf of

Dr. Reginald B. Kogbara 

Academic Editor

PLOS ONE